# Automated workflow for the cell cycle analysis of (non-)adherent cells using a machine learning approach

Kourosh Hayatigolkhatmi[1†], Chiara Soriani[1†], Emanuel Soda[1†], Elena Ceccacci[1], Oualid El Menna[1], Sebastiano Peri[1], Ivan Negrelli[2], Giacomo Bertolini[2], Gian Martino Franchi[2], Roberta Carbone[2], Saverio Minucci[1,3]*, Simona Rodighiero[1]*

[1]Department of Experimental Oncology, European Institute of Oncology-IRCCS, Milan, Italy; [2]Tethis S.p.A., Milan, Italy; [3]Department of Oncology and Hemato-Oncology, University of Milan, Milan, Italy

*For correspondence:
saverio.minucci@ieo.it (SM);
simona.rodighiero@ieo.it (SR)

†These authors contributed equally to this work

Competing interest: The authors declare that no competing interests exist.

## eLife assessment

This article presents a **valuable** automated method to track individual mammalian cells as they progress through the cell cycle using the FUCCI system. The authors have developed a technique for analyzing cells that grow in suspension and used their method to look at different tumor cell lines that grow in suspension and determine the effect of drugs that directly affect the cell cycle. They show **solid** evidence that the method can be applied to both adherent and non-adherent cell lines. This article will be of interest to cell biologists investigating cell cycle effects.

**Abstract** Understanding the cell cycle at the single-cell level is crucial for cellular biology and cancer research. While current methods using fluorescent markers have improved the study of adherent cells, non-adherent cells remain challenging. In this study, we addressed this gap by combining a specialized surface to enhance cell attachment, the FUCCI(CA)2 sensor, an automated image analysis pipeline, and a custom machine learning algorithm. This approach enabled precise measurement of cell cycle phase durations in non-adherent cells. This method was validated in acute myeloid leukemia cell lines NB4 and Kasumi-1, which have unique cell cycle characteristics, and we tested the impact of cell cycle-modulating drugs on NB4 cells. Our cell cycle analysis system, which is also compatible with adherent cells, is fully automated and freely available, providing detailed insights from hundreds of cells under various conditions. This report presents a valuable tool for advancing cancer research and drug development by enabling comprehensive, automated cell cycle analysis in both adherent and non-adherent cells.

## Introduction

Cell cycle dynamics coordinate cellular division and proliferation through regulating the different cell cycle phases. Dysregulation in these processes is a hallmark of malignancies such as human cancer, where aberrant activities in cyclin-dependent kinases (CDKs), cyclins, and CDK inhibitors often drive uncontrolled proliferation. Consequently, targeting cell cycle components has emerged as a pivotal therapeutic strategy, especially crucial in preclinical drug evaluation (*Malumbres and Barbacid, 2009*; *Khan and Wang, 2022*).

Traditional methods for assessing cell cycle dynamics have been largely dependent on the quantification of DNA content through flow or image cytometry, providing a static snapshot of cell populations

**eLife digest** Cells regularly grow and divide through a process called the cell cycle. It includes rest periods where no growth or division occurs. When the cells are ready to divide, they duplicate their DNA, so each new cell gets a complete set of instructions. Finally, the cell splits into two new cells through a process called cytokinesis. This whole process can take hours or days to complete, depending on the cell type. Many things can go wrong during these processes, impairing healing or causing tumor formation. Learning more about these processes could help scientists better understand healing and diseases like cancer.

Emerging imaging and data analysis tools allow scientists to observe cell-growth processes and watch errors as they occur. But, doing so requires sophisticated equipment and can be time and labor-intensive. Especially, if scientists are trying to track the cell cycle in a large number of cells. It can also be challenging to track free-moving cells, like blood or immune cells. New tools and techniques are needed to help scientists overcome these challenges.

Hayatigolkhatmi, Soriani, Soda et al. developed a method in which a sticky surface is used to grow blood cancer cells that allows them to observe the cell cycle in large numbers of the cells at the same time. In the experiments, blood cancer cells were grown on a nano-material-coated surface that kept the usually free-floating cells still. The team compared gene expression in the cells before and after raising them on the surface to confirm that confining the cells did not alter their gene expression or disrupt their normal life cycle. Then, the researchers developed machine learning software that monitors the cell cycle in hundreds of individual cells, quantifies cell cycle phases and analyzes data with minimal human intervention. Usually, it would take a scientist 40-50 hours to oversee the cell cycle in a single experimental condition. This time was reduced to approximately 2 hours for a complete experiment using their pipeline. Finally, they validated their tools by monitoring different types of cancer cells under various treatment conditions.

The tools developed by Hayatigolkhatmi, Soriani, Soda et al. provide researchers with a fast, easy and cost-effective tool for studying the cell cycle. It could help scientists study early development and how cells differentiate, grow or age. It could also be helpful for scientists studying cancer and how to treat it or scientists studying the healing process.

---

in various cycle phases (*Furia et al., 2013*; *Ligasová et al., 2023*). Although valuable, these techniques fall short in capturing intra-population variability and require additional protein markers for precise phase determination (*Ligasová et al., 2023*; *Rieger, 2022*).

Methods utilizing cells expressing fluorescently labeled reporters and time-lapse microscopy can discriminate cell cycle phases at the level of individual cells, thereby offering valuable insights into the variability of cell cycle and cell cycle phase durations within the overall cell population (*Chao et al., 2019*; *Hiratsuka and Komatsu, 2019*).

Advanced imaging methods, such as time-lapse microscopy coupled with fluorescently tagged reporters, have shown promise in detailing cell cycle dynamics at the single-cell level. Technologies like the Fluorescent Ubiquitination-based Cell Cycle Indicator (FUCCI) have been employed for this, effectively demarcating cell cycle phases through color-coding (*Sakaue-Sawano et al., 2017*). FUCCI(CA)2 express the hCdt1(1/100) fused to mCherry fluorescent protein and hGem(1/110) fused to mVenus, generating a clear and distinct tricolor demarcation, separating G1 (red), S (green), and G2/M phases (yellow) (*Sakaue-Sawano et al., 2017*).

This dynamic insight is particularly crucial in the context of acute myeloid leukemia (AML), where chromosomal translocations generate fusion genes that disrupt cellular differentiation programs and drive proliferation (*Alcalay et al., 2001*).

The described experimental setup utilizes nanostructured titanium oxide-coated multiwell plates relying on the technology used in the commercially available Smart BioSurface (SBS) slides (*Krol et al., 2021*). Such technology should hypothetically enable us to immobilize non-adherent cells for extended imaging durations. To overcome the limitations of manual data analysis, we introduce an automated image analysis pipeline for time-lapse videos of AML cell lines, exploiting a FUCCI-based probe for visualization. Our data analysis approach combines custom image processing, TrackMate-based cell tracking, and machine learning-based track filtering, thereby automating the entire data analysis workflow.

In summary, we present a comprehensive, experimental protocol for cell cycle analysis in adherent and non-adherent cells (summarized in *Box 1*). The approach leverages routine imaging technologies and advanced data analyses, enhancing the precision and efficiency of drug screening protocols in oncological research.

## Results

### Modified conditions enable AML cells to adhere to the substrate feasible for live-cell imaging

Live imaging and tracking of non-adherent cells, when multiple positions should be acquired, is challenging due to their high propensity to mechanical perturbations.

To conduct long-time imaging of AML cells, we exploit the combined action of the SBS and the partial immobilization effect of methylcellulose (MC). We were able to image and track AML cells up to 72 hr when 20% complete medium was added to 80% MC and applied on AML cells previously adhered to the SBS (*Video 1*).

Two different AML cell lines, NB4 and Kasumi-1, were equipped with the FUCCI(CA)2 technology and chosen as the study models. NB4 cells have relatively faster doubling time in comparison to Kasumi-1 cells (*Skopek et al., 2023*). Hence, NB4 cells were treated with vehicle or CDK inhibitors, being compared to the naturally slow-cycling counterparts, Kasumi-1 cells, by time-lapse imaging (*Figure 1A*, *Figure 1—figure supplement 1*). As a result, we were able to follow single cells and the manual annotation of cell cycle phases according to the color of the red and green merged images in *Figure 1A*, qualitatively confirmed the impact of CDK inhibitors. Furthermore, the expected difference in cell cycle progression of NB4 vs Kasumi-1 cells was evident.

### Image processing facilitates the cell tracking and profiling of cell cycle phases

The original time-lapse images were composed by red and green channels, detecting respectively the mCherry and mVenus markers of the FUCCI(CA)2 indicator. During the experiment, each cell alternatively switches between the expression of these two markers as it goes through the different cell cycle phases, causing the lack of a single fluorescence channel suitable for tracking. Furthermore, the automatic profiling of cell cycle phases becomes cumbersome when dealing with two distinct channels that in the end give rise to two independent fluorescence time series. We set up an image processing pipeline utilizing the open-source software Fiji (*Schindelin et al., 2012*; NIH, version 2.14.0/1.54f) to transform the original images into a dataset optimized for the subsequent steps of tracking and cell cycle phase assignment. We represented the color changes that occur during the cell cycle with the Hue scale, as described in *Fujimoto et al., 2020*. To achieve this, as represented in *Figure 1B*, stacks of red and green channels were overlaid and converted into an RGB stack and then transformed into an HSB stack (hue, saturation, and brightness channels). The hue channel was retained for the assignment of cell cycle phases, while the brightness channel was used as tracking reference. These two were merged to the red and green fluorescence channels to form the final stack used for the tracking process.

In the tracking analysis, the Fiji plugin TrackMate (*Tinevez et al., 2017*) was employed, adapting the example script available on the dedicated TrackMate website (https://imagej.net/plugins/trackmate/scripting/scripting), to ensure the automation of the tracking step. The related parameters were selected and tuned according to each experiment, as well as the proper filters to discard uninformative tracks. The output consisted of a table with cells associated to selected tracks and their corresponding numerical features in each time frame. Key features included the mean fluorescence intensity in the red and green channels, as well as the mean intensity of the hue value.

The whole image analysis pipeline successfully recovered a unique tracking channel and effectively mapped the alternating red and green curves (*Figure 1C*, left panel) into a single time series, as depicted in *Figure 1C*, right panel.

## Box 1. Experimental and analysis workflow.

**Sample preparation (in-suspension cells):**
The cells of interest should be at optimum viability and physiological conditions prior to the mounting process on the SBS-multiwell plates. This step requires 1–2 hr of manual handling and is divided into seven steps as follows:

I.  Consider 250,000 cells for one well of a standard 12-well SBS-multiwell. Wash the cells three times in sterile phosphate-buffered saline (PBS).

Note 1: PBS should be at room temperature (RT).
Note 2: The media and PBS should not be vacuumed during the process. It is crucial to make sure the cells won't get dried at any point of the process.
Note 3: The mentioned number of cells is for cells with an average size of 10–20 μm in diameter and a doubling time of 20–30 hr. The number of cells with different properties should be optimized accordingly.

I.  Resuspend the cells in 1000 μl PBS and gently mix, achieving a homogeneous cell suspension.

II.  Load the suspension slowly from one side of each well, covering all the surface.

III.  Let the cells mount for 20–30 min at RT, followed by 20–30 min at 37°C.

Note 4: Avoid moving or shaking the plate or the surface underneath.

I.  Remove the PBS with pipette (we suggest not to use the vacuum and not to tilt the well).

II.  Wash the wells two times with enough volume of media without fetal bovine serum (FBS) (we suggest not to use the vacuum and not to tilt the well).

Note 5: Any protein contamination, including FBS, can easily perturb adhesion of the cells on the titanium-oxide particles.
Note 6: Load and aspirate slowly from the side of the wells, keeping the pipette at 45° angle.

I.  Load enough methylcellulose gently from one side of the well and proceed to the next step.

Note 7: In case any treatments are needed during the time-lapse experiment, the desired agents to can be added to the methylcellulose compartment prior to loading on the wells.

**Sample preparation (adherent and semi-adherent cells):**
In case the cells of interest are (semi-)adherent cells, plate the cells at optimum confluency (based on the planned time-lapse duration and cell properties) on glass-bottom dishes and proceed to the next step.

**Live-cell imaging**

I.  Pre-heat the microscope incubator at 37°C and 5% of $CO_2$ for standard cell culture. Otherwise, tune the temperature and humidity controller according to the ideal cell viability conditions.

II.  Mount the sample on the microscope stage and choose the imaging parameters depending on the cell type.

Note 8: We suggest setting the objective and image binning to represent each cell with at least 15 pixels in diameter.
Note 9: We suggest adjusting the excitation intensity and exposure time for the detection of mCherry and mVenus signal to have a difference between maximum and minimum gray value of at least 1000 units (for 16-bit camera).

III.  Select the desired total duration of the experiment and the time-frame parameter, considering the cell-type viability upon illumination and the speed of cells movement.

Note 10: For tracking purposes, it is convenient to start with a time interval of 30 min and evaluate the outcome of the time lapse, adjusting eventually the described parameter in the following experiments.

**Image processing**

I.   Open the preprocessing macro (Image_processing_HSB_v1.ijm) in your Fiji.

Note 11: the execution of the macro requires the prior installation of the Basic plugin (https://github.com/marrlab/BaSiC; **Peng, 2022**) for flat-field correction.

II.  Set the proper image filters and background subtraction processing for red and green channels at lines 58–78 and 85–94.

III. Choose the maximum values of the brightness and contrast for both red and green channels and set the found values at lines 116 (red) and 118 (green).

IV.  Set the proper value of the Top Hat filter on the brightness channel (HSB stack) at line 127.

V.   Set the correct position of red and green channels, depending on the experiment, at lines 173–174.

VI.  Run the macro from the Fiji script editor.

VII. Set in the dialog window the input directory (where original data are stored), the output directory (where you want to store processed images), and the original file format.

**Tracking analysis**

I.   Open the TrackMate script in the Fiji macro editor.

II.  Adjust the parameters settings (from lines 39–58) according to the experiment.

III. Run the script from the Fiji script editor.

IV.  Set in the dialog window the input directory (where the images processed with the previous pipeline are stored) and the output directory (where you want to store the results table).

**Data analysis**

**Sample/condition demultiplexing**

I.   Read the list of files containing tracking information for each cell. This can be done by looping through the files and reading into R using the function read.delim().

II.  Merge all the tables into a single table using the filename to create a column for the demultiplexing of the condition.

**Missing frame imputation**

I.   Use the na_ma() function to perform imputation. This function lets the user choose among different weighting strategies; we suggest trying the 'exponential' or the 'simple', as well as trying different values for k (dimension of the moving average window) and maxgap (maximum number of successive NA values).

**Data smoothing**

I.   Use the function smooth.spline() to remove noise from the time series and enhance the trend. We suggest trying different values for lambda value starting with a very low one such as [1e-05, 1e-04, 1e-03].

**Data normalization**

I.   Use the function normalize_vector() to make the curves comparable to each other.

**Feature extraction**

I. Use the function tk_tsfeatures () to extract features from the channel intensity; different features can be specified to be extracted such as

 a. The number of times the time series crosses its median.

 b. Autocorrelation of the time series.

 c. Autocorrelation of the first-/second-differentiated time series.

 d. Spectral entropy.

 e. Stability and lumpiness on a tilled version of the time series.

**Traceability assessment**

I. Use the function predict() passing the table and the pre-trained model to predict whether a track is traceable or not. The output is a table containing the predicted class as well as the predicted probability of being part of a specific class.

II. The predicted probability can be used to further filter the predicted traceable cells.

**Cell phase assignment**

I. The cell cycle phase quantification is performed on the cells predicted as traceable; this is done by setting two thresholds on the Hue intensities. We suggest trying different values according to the original Hue intensity reached by the fluorescence observed.

II. The first round of phase assignment divides the tracks into:

 a. G1

 b. S

 c. G2/M

III. By looping through each track, a second round of phase assignment is performed to reassign frames wrongly classified as G2/M that occur before S. Two different strategies can be followed:

 a. Reassign the G2/M frames to S.

 b. Reassign the G2/M frames to a new G1/S phase.

**Cell phase quantification**

I. Group your data using the group_by() function using the Track_ID and Condition columns as grouping variables then use the function add_count() to count the number of frames in each condition for each track.

II. Divide the number obtained by the number of frames acquired in an hour to obtain the time spent in each phase (in hours units).

III. Sum the time spent in each phase to obtain the total cell cycle duration.

## Incorrect track filtering by the machine learning model efficiently automates data cleaning and cell cycle phase assignment steps

The files generated by the TrackMate pipeline were imported into R (*R Development Core Team, 2021*) for the demultiplexing, filtering, and data analysis. To address missing frames, a data imputation process was implemented, aimed at recovering instances with up to five consecutive missing values. This data imputation was executed using an exponential weighted moving average (EWMA) technique, employing a window size of 3, thereby encompassing six observations (three preceding, three succeeding).

A smoothing spline approach was individually applied to the green, red, and hue channels, utilizing a regularization parameter (lambda) of 0.0001. A min-max scaling to transform value into the range [0, 1] was applied to make the fluorescence intensity and hue scale comparable among channels

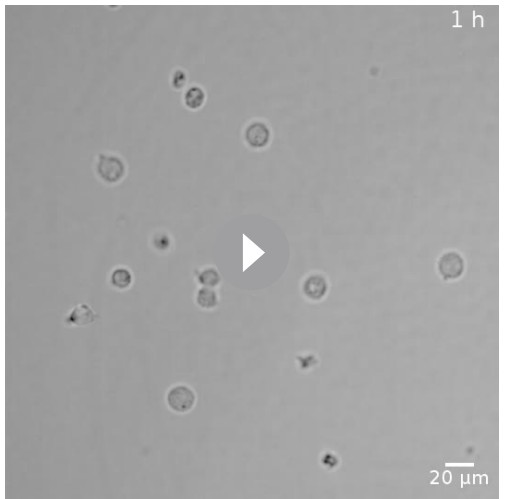

**Video 1.** Representative video of NB4 cells seeded on Smart BioSurface (SBS) + methylcellulose (MC).

https://elifesciences.org/articles/94689/figures#video1

and samples. For the creation of the random forest model, a training pipeline was set up, as described in *Figure 2A*. In accordance with *Figure 2B*, feature extraction was carried out on the green and red channels to derive temporal descriptors of the data.

Employing these extracted features, the trained random forest model was applied to discriminate between traceable and untraceable cells. Subsequently, only the cells predicted as traceable by the model were retained for subsequent analyses. Here are reported the obtained number of cells for each condition, merging all the experiments performed:

- NB4 DMSO: 410 cells (three experiments) out of 1862 tracked cells.
- NB4 palbociclib 50 nM: 328 cells (three experiments) out of 2881 tracked cells.
- NB4 PF-06873600 50 nM: 206 cells (one experiment) out of 1102 tracked cells.
- NB4 ribociclib 50 nM: 119 cells (one experiment) out of 784 tracked cells.
- Kasumi-1 untreated: 119 cells (one experiment) out of 1604 tracked cells.
- MDA-MB-231 untreated: 1116 (one experiment) out of 3204 tracked cells.

By using the hue channel, each track was partitioned into its cell cycle component using the following thresholds:

- G1: hue ≥ 0 and hue <0.65.
- S: hue ≥ 0.85.
- G2/M: hue ≥ 0.65 and hue ≤ 0.85.

Additionally, a refinement step involving cell reassignment was conducted to identify instances where the G1 to S phase transition was labeled as G2/M in frames preceding the S phase. The outcome of the analysis is visually depicted in *Figure 3A and B*.

## Evaluation of tracks' selection performances confirms the efficiency of the machine learning model

We carried out a validation of the performances of the whole pipeline in selecting 'traceable' tracks. Specifically, we extracted from one experiment the ID of the tracks that belonged to the vehicle (DMSO)-treated condition identified as 'traceable' by the ML algorithm. We then validated these tracks by visual inspection of the corresponding images, checking for eventual tracker errors. We created a scoring metric defined as follows:

$$Score = \frac{N_{Tracks\,(images)}}{N_{Tracks\,(ML\,algorithm)}}$$

$N_{Tracks}$ *(images)* = the number of tracks selected as *traceable* by the algorithm that were good also in images.

$N_{Tracks}$ *(ML algorithm)* = the number of tracks selected as *traceable* by the algorithm.

We checked up to 102 tracks that underwent whole cell cycle profiling by the analysis pipeline and found that, among these, 63 were also consistent with images (score = 0.62). We then decided to select tracks identified as traceable with a probability greater than 0.70, gaining a final score of 0.74 (57 tracks consistent with images over 77 tracks).

## The workflow allows the identification of the G1 to S phase transition

The empirical observation of the superimposition of red and green fluorescence signals during the G1 to S transition makes the classification of the cell in one of the two dual states challenging. Cells

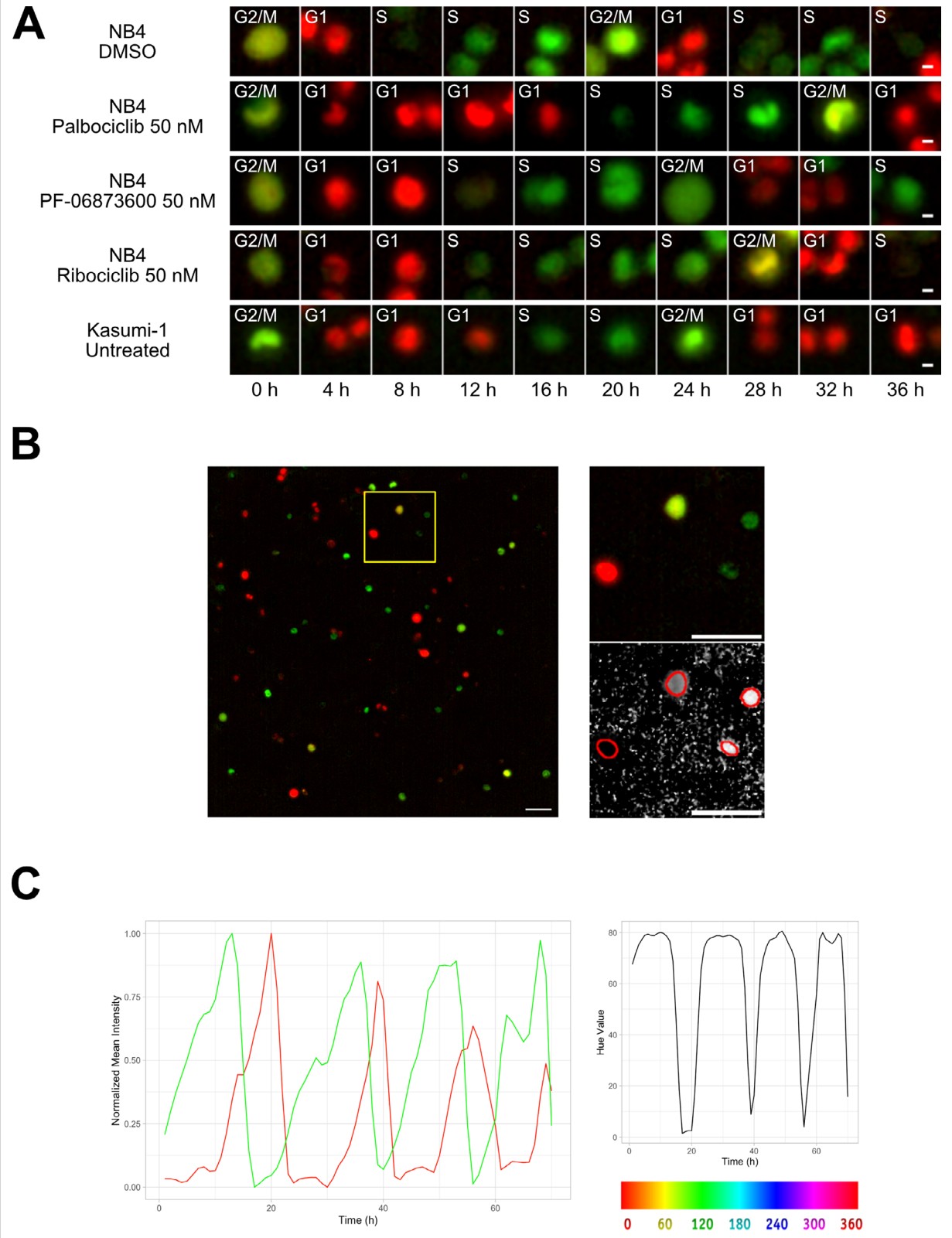

**Figure 1.** Cell tracking. (**A**) Examples from Kasumi-1 cell and NB4 cell lines (upon the different conditions: DMSO, palbociclib 50 nM, PF-0606873600 50 nM, ribociclib 50 nM, untreated), showing a tracked cell that explores the different cell cycle phases (scale bar is 5 μm). (**B**) Fluorescence images with R and G channels were processed in Fiji to create an HSB stack, from which the brightness and hue channel were extracted to be used respectively as tracking channel and cell phase identification channel (scale bar is 50 μm). (**C**) Plot curves generated using the data extracted from the TrackMate

*Figure 1 continued on next page*

*Figure 1 continued*

script execution. The variations over time of mCherry (red curve) and mVenus (green curve) fluorescence intensities, normalized between minimum and maximum values (left plot), and of the hue scale (right plot) of a single NB4 cell in DMSO condition are shown.

The online version of this article includes the following figure supplement(s) for figure 1:

**Figure supplement 1.** Cell cycle phases in treated NB4 cells.

often exhibit a brief gap in fluorescence during this transition, as illustrated in the first image categorized as S phase in both *Figure 1A* and Figure S1. This apparent lack of fluorescence occurs because mCherry fluorescence rapidly decreases to near zero at the start of the S phase, while mVenus fluorescence begins to increase slowly (*Figure 1C*). Although seemingly non-fluorescent, this initial S phase is marked by the coexistence of both mCherry and mVenus fluorescence, producing a yellow hue in the scale before the S phase, labeled as the G1 to S transition (G1/S) by the refinement step in our model. Due to the lower intensities of mCherry and mVenus during this transitional phase, compared to other cell cycle phases, the yellow fluorescence is barely visible in fluorescence images where brightness and contrast settings are dominated by the bright G1 red and S green fluorescence.

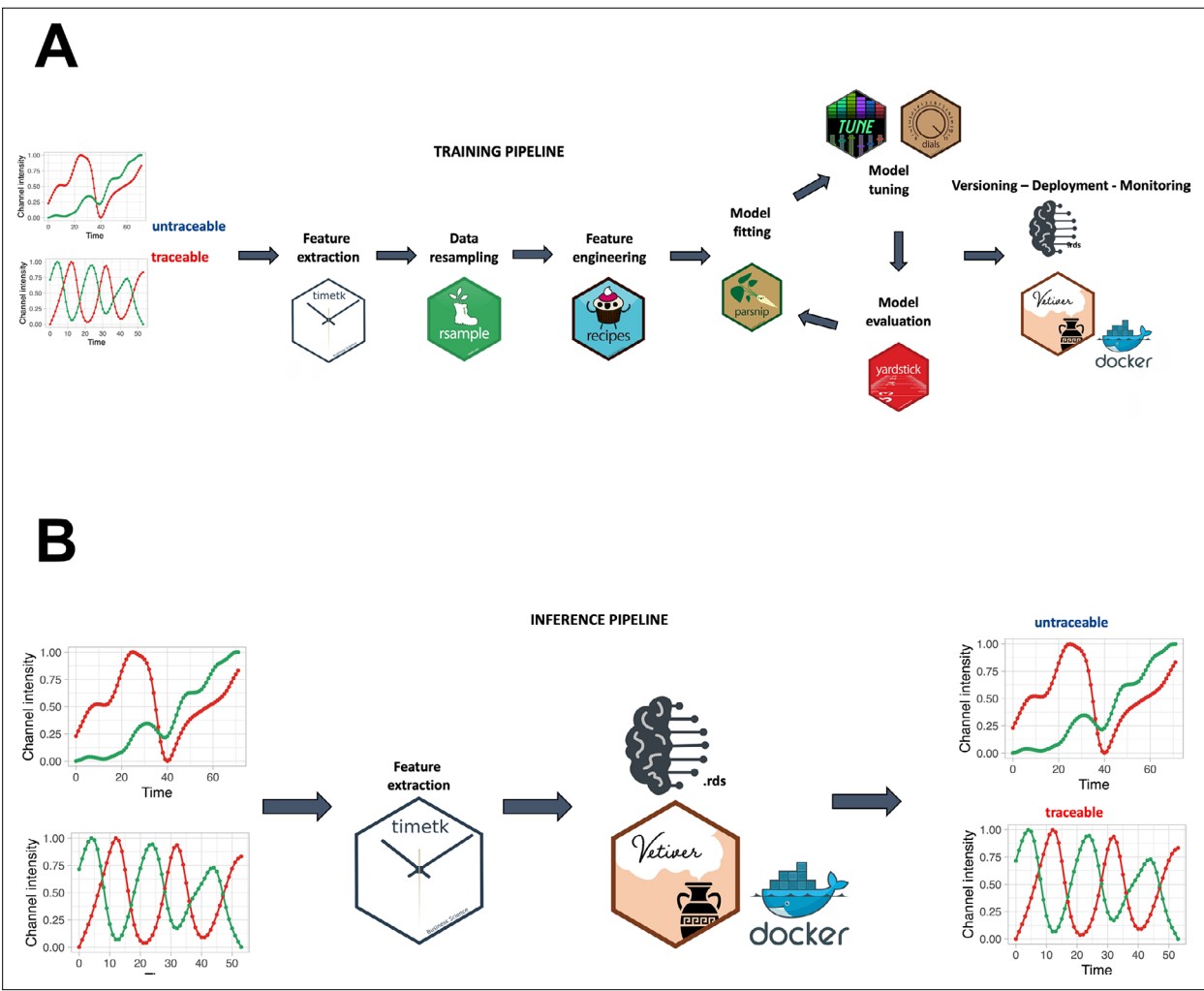

**Figure 2.** Training and inference pipelines. (**A**) The machine learning pipeline followed to create the quality model. Using timetk, time series-associated features are extracted from the list of manually annotated tracks. A random forest model is then trained to predict whether a track is cycling or not. (**B**) An unannotated track can be fed to the model to predict whether it is cycling or not.

The online version of this article includes the following figure supplement(s) for figure 2:

**Figure supplement 1.** Feature distribution and model performance.

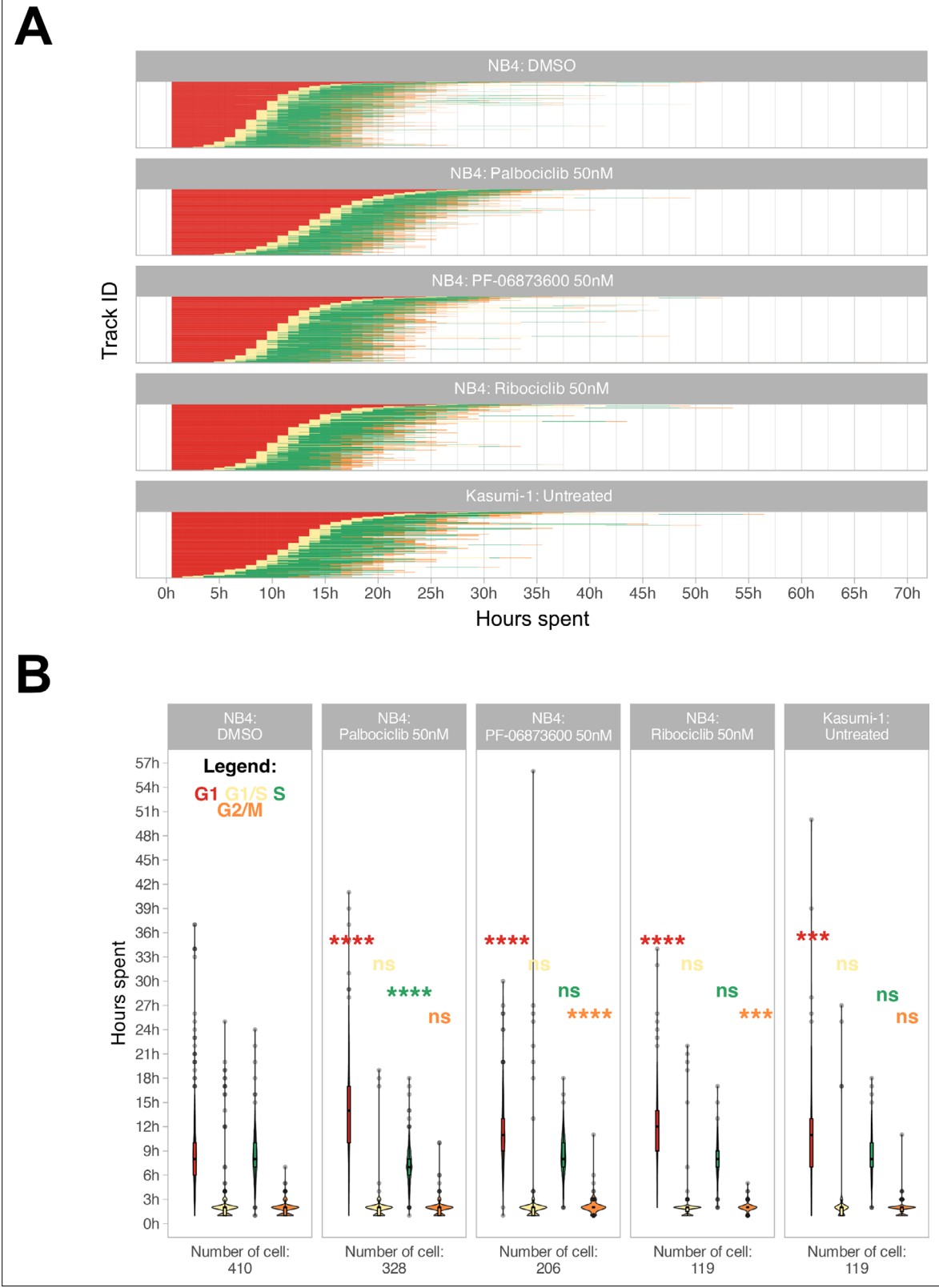

**Figure 3.** Cell cycle phase assignment. (**A**) Waterfall plot of treated and untreated acute myeloid leukemia (AML) cells. Each row corresponds to a single cell. (**B**) Boxplot of the cell phase duration of the first cell cycle, as obtained by the pipeline, without excluding of the outliers. The asterisks are the adjusted significance level given by Wilcox test of the sample for each over the NB4 DMSO for each phase according to the color (ns: p>0.05, *p≤0.05, **p≤0.01, ***p≤0.001, ****p≤0.0001).

*Figure 3 continued on next page*

*Figure 3 continued*

The online version of this article includes the following figure supplement(s) for figure 3:

**Figure supplement 1.** Cell cycle phase assignment in MDA-MB-231 cells.

**Figure supplement 2.** Transcriptome correlation analysis of acute myeloid leukemia (AML) cells seeded on Smart BioSurface (SBS).

To evaluate the relative expression of the two fluorescent probes during this transition, we conducted flow cytometry and confocal experiments on Kasumi-1 and MDA-MB-231 cells expressing the FUCCI(CA)2 probes. In these experiments, the S phase was labeled by a pulse of EdU and the DNA by DAPI or Hoechst 33342 (*Figure 4*, *Figure 4—figure supplement 1*). *Figure 4A* shows cell cycle profiling based on EdU vs. DAPI can successfully discriminate G1, S, and G2/M phases, with the S phase being subdivided into early, mid, and late S phases (*Figure 4A*, *Figure 4—figure supplement 1A*). The proportion of cells in each phase is comparable to the cell cycle profiling of the same cells by FUCCI(CA)2 (*Figure 4B*, *Figure 4—figure supplement 1B*). Moreover, the temporal transition through the S phase discovered by EdU correlates with the mVenus emission levels detected by FUCCI(CA)2 alone (*Figure 4B*, *Figure 4—figure supplement 1B*). This is evident by the accumulation of the EdU-early S phase cells on the left side of FUCCI(CA)2 S phase cells (low mVenus emission). On the contrary, the EdU-late S phase cells tend to accumulate on the right side of the FUCCI(CA)2 S phase cells (high mVenus emission) (*Figure 4B*, *Figure 4—figure supplement 1B*). This finding is in line with the earlier observation of the relatively rapid loss of mCherry at late G1 and the gradual increase in mVenus emission throughout the S phase (*Figure 1C*). These observations suggest the possibility that FUCCI(CA)2 alone can discriminate early vs. late S phase cells. However, this aspect can be exploited in cell populations with necessarily similar levels of FUCCI(CA)2 expression.

Interestingly, from *Figure 4B*, it is evident that the EdU-early S phase cells can express mCherry and mVenus at different levels. *Figure 4C* shows segmented nuclei of cells at the initial S phase (arrows), characterized by dim EdU staining and DNA amount only slightly higher than 2N (not shown), where the dim green and red fluorescence overlaid determined a predominant red (when the intensity of mCherry is slightly higher than the intensity of mVenus), yellow (when the intensity of mCherry and mVenus are similar), or predominant green (when the intensity of mVenus is slightly higher than the intensity of mCherry) color in the merged image. The same phenomenon was observed in MDA-MB-231 cells (*Figure 4—figure supplement 1*). We examined this phenomenon in three of the previously shown cells – Kasumi-1 and PF-06873600-treated NB4 in *Figure 1A* and MDA-MB-231 in *Figure 3—figure supplement 1C* – where all the available frames were visually evaluated. The G1 to S transition was once captured as predominantly red (mCherry > mVenus, *Figure 4D*), once as yellow (mCherry ≈ mVenus, *Figure 4D*), and once as green (mVenus > mCherry; *Figure 4—figure supplement 1D*).

## The method can quantify the cell cycle progression and does not impact the transcriptome over long time periods

We were able to accurately quantify the cell cycle progression of NB4 and Kasumi-1 cell lines over 72 hr at single-cell resolution and, as represented in *Figure 3A and B*, cell cycle differences of NB4 and Kasumi-1 cells were quantified. Moreover, the method was able to discriminate cell cycle differences of leukemic cells in presence of various suboptimal doses of different CDK inhibitors. These CDK inhibitors are potent inhibitors of CDKs2/4/6, which mainly regulate the G1 to S transition (*Fassl et al., 2022*). Hence, we expected the administration of nanomolar concentrations of these agents to lengthen mainly the G1 phase with less impact on the other phases. Our live-cell imaging of the relatively fast-cycling NB4 cells in the presence of these drugs at such spectrum of concentrations affirmed the significant G1-prolongation in these cells (*Figure 3B* and *Videos 2–4*). The total average duration of one full cell cycle was quantified as 21.5 hr ± 6.5 hr (mean ± standard deviation) for NB4 cells and 24.0 hr ±7.8 hr for Kasumi-1 cells. For instance, administration of 50 nM palbociclib extended mainly the G1 phase of the NB4 cells by about 5 hr (from 9.1 hr ± 5.1 hr to 14.2 hr ± 5.7 hr in vehicle-treated and palbociclib-treated NB4 cells, respectively).

It was demonstrated that the nanostructured surfaces can promote changes in the cellular protein expression profile (*Schulte et al., 2016*). We thus questioned the possible impact of the SBS on gene expression profile of the AML cells. We made a comparison of transcriptome of the cells pre- and post-imaging by performing RNA-seq investigations. As represented in *Figure 3—figure supplement*

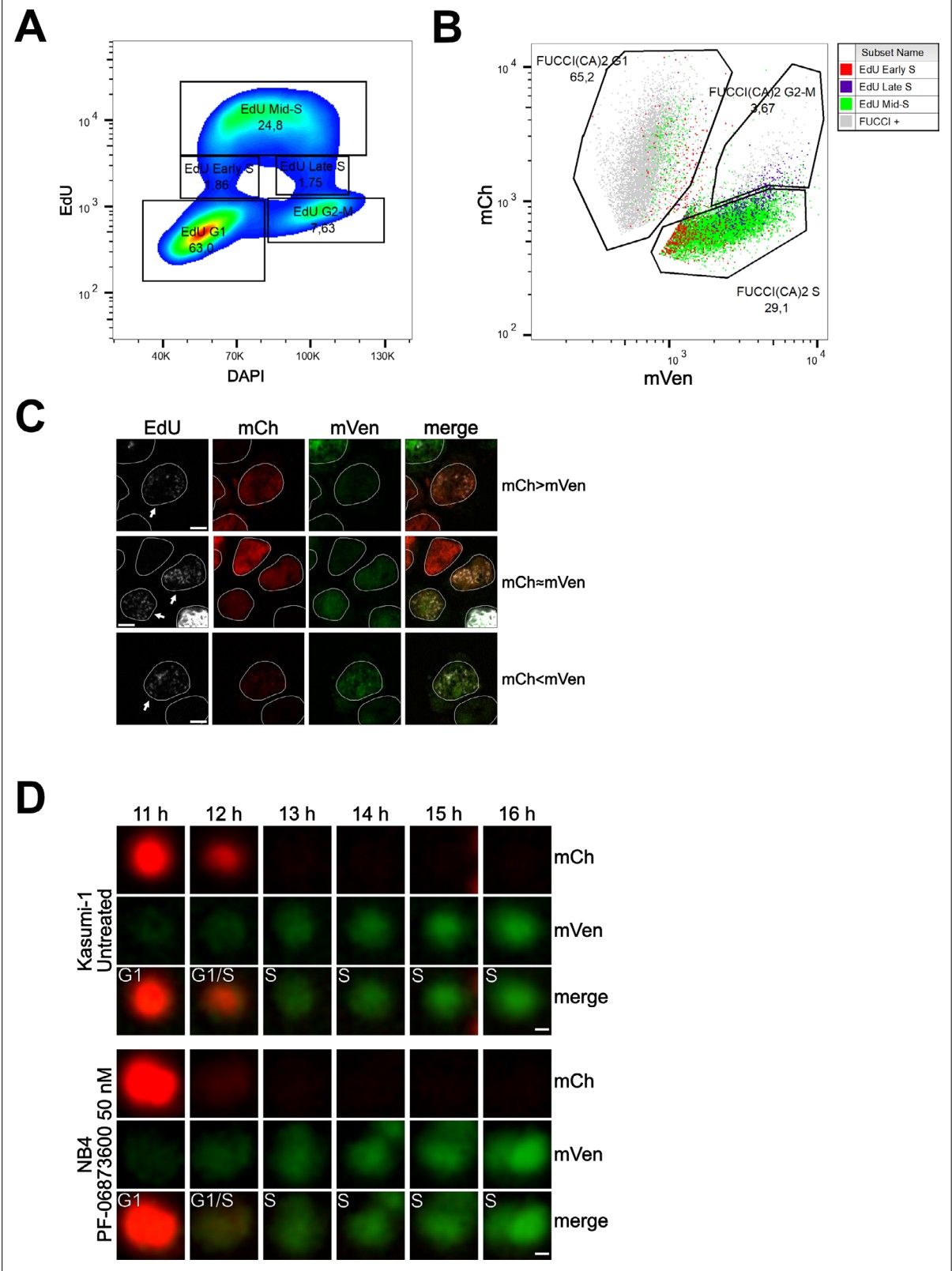

**Figure 4.** Cell cycle profiling and G1 to S transition analysis. (**A**) Density plot showing the cell cycle profiling of FUCCI(CA)2-Kasumi-1 cells based on EdU vs DAPI. (**B**) Scatter plot of the same cells as in (**A**) showing the cell cycle profiling based on FUCCI(CA)2. Cells identified to be in early, mid, and late S phase by EdU are highlighted in red, green, and purple, respectively. Due to the chosen gates and the 2 hr EdU pulse, a small percentage of EdU-positive cells leaked into the G1 and G2-M FUCCI(CA)2-defined groups. (**C**) Representative images of nuclei during the G1 to S transition are indicated

*Figure 4 continued on next page*

*Figure 4 continued*

by arrows. Depending on the exact time after S phase initiation, the overlay of the mCherry and mVenus channels can show a red prevalence (first row), a yellow color (middle row), or a green prevalence (third row). The scale bar represents 5 µm. (**D**) The frames, including the transition from G1 to S phase of the untreated Kasumi-1 cell and the NB4 PF-06873600-treated cell shown in *Figure 1A*, are presented from the 11 hr to the 16 hr frame. In these image sequences, the G1/S label was assigned to the first frame showing dim but detectable mVenus fluorescence and dim but still detectable mCherry fluorescence. The scale bar represents 2 µm.

The online version of this article includes the following figure supplement(s) for figure 4:

**Figure supplement 1.** Cell cycle profiling and G1 to S transition analysis in MDA-MB-231 cells.

*2*, we did not observe any meaningful transcriptomic alterations throughout the process. Hence, we conclude that the protocol is feasible for biological investigations without a significant effect on the transcriptomic profile of the cells.

## Discussion

In the last few years, efforts have been made to simplify the cell cycle assessment relying on FUCCI technology, resulting in the development and public availability of software tools and ImageJ plugins (*Roccio et al., 2013*; *Koh et al., 2017*; *Ghannoum et al., 2021*; *Taïeb et al., 2022*). However, all these methods require human intervention at different points in the analysis workflow, such as the initial selection of cells for analysis (*Taïeb et al., 2022*) or the manual correction of inaccurate tracks during the analysis (*Roccio et al., 2013*; *Koh et al., 2017*; *Ghannoum et al., 2021*; *Taïeb et al., 2022*). In this report, we describe a complete protocol for the cell cycle analysis of adherent and non-adherent cells expressing the FUCCI(CA)2 technology in a fully automated manner. The complete experimental workflow is explained in *Box 1* and illustrated in *Figure 5*. It was applied to the analysis of the cell cycle phases of two different AML cell lines, NB4 and Kasumi-1, which have different durations. Hence, detecting this difference was a reliable initial verification strategy for the protocol's efficiency. As a second quality check step, we decided to treat the relatively fast-cycling NB4 cells with suboptimal concentrations of three different CDK inhibitors (palbociclib, PF-06873600, and ribociclib). These CDK inhibitors have different inhibitory impact on various CDKs (*Freeman-Cook et al., 2021*; *Fassl et al., 2022*). The effect of the three CDK inhibitors on the cell cycle duration of NB4 cells was evaluated. Up to about 400 cells in one single experimental condition were quantified, for up to 12 different conditions in a single experiment. This entire analysis process took approximately 2 hr of human involvement, for a total execution time that ranges from 12 to 48 hr, depending on dataset size, that is commissioned

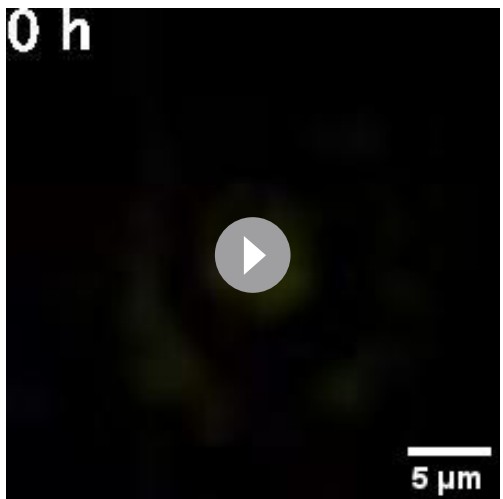

**Video 2.** Representative videos of a NB4 cell in control condition (DMSO) tracked with TrackMate expressing the FUCCI(CA)2 probe, showing the sequence of color changes (1 frame/hr).

https://elifesciences.org/articles/94689/figures#video2

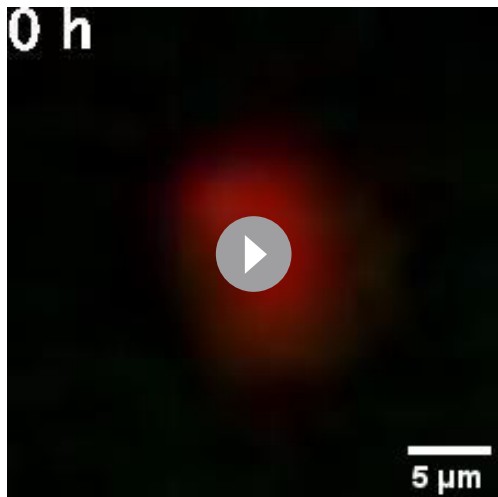

**Video 3.** Representative video of a Kasumi-1 cell tracked with TrackMate expressing the FUCCI(CA)2 probe, showing the sequence of color changes (1 frame/hr).

https://elifesciences.org/articles/94689/figures#video3

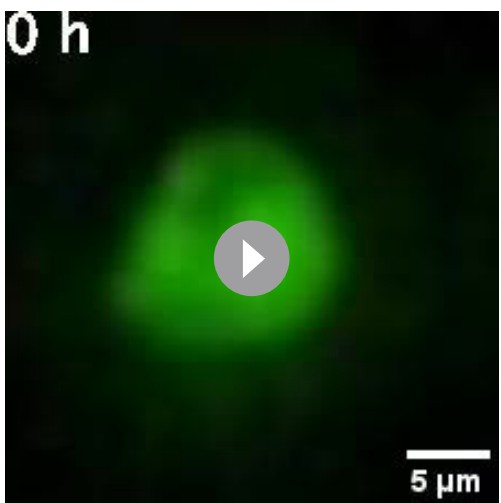

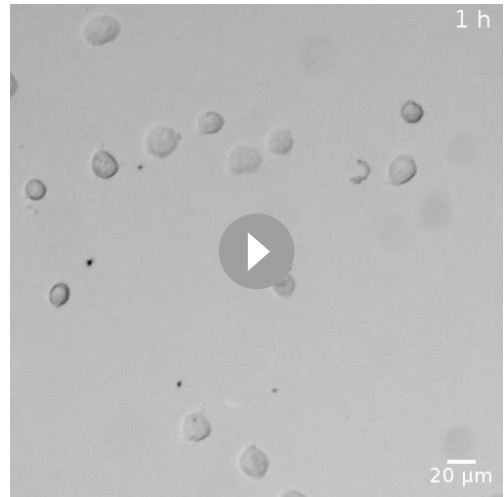

**Video 4.** Representative videos of NB4 cell treated with palbociclib 50 nM and tracked with TrackMate expressing the FUCCI(CA)2 probe, showing the sequence of color changes (1 frame/hr).
https://elifesciences.org/articles/94689/figures#video4

**Video 5.** Representative video of NB4 cells seeded on Smart BioSurface (SBS) without methylcellulose (MC).
https://elifesciences.org/articles/94689/figures#video5

to a machine (*Figure 5*). It should be taken into consideration that the manual correction of incorrect tracks takes approximately 2 hr per field of view (FOV). Hence, in the reported experiments of this study, the analysis of a single well would take about 40–50 hr of manual work. This amount of time would make such analysis unfeasible. Not only the suggested protocol makes these experiments doable, but also the experimental set-up can be scaled up according to the experimental needs. We successfully emphasized alterations of few hours in the duration of cell cycle phases when NB4 cells were subjected to exceedingly low concentrations of inhibitors. Moreover, the set-up pipeline makes possible the quantification of potentially thousands of cells in an automated fashion relying on the crucial contribution of a machine learning algorithm.

Various approaches have been documented for imaging non-adherent cells. These range from creating custom supports to confine cell movement within a restricted region enabling extended imaging periods of up to 40 hr (*Day et al., 2009*), to partially immobilizing cells using substances like gelatin (*Ritter, 2020*) or low-melting-point agarose (*Strong and Daniels, 2017*).

The capability to track non-adherent cells for up to 72 hr (almost equivalent to three complete cell cycles in NB4 cells) was achieved by employing SBS-coated glass in combination with the addition of MC to the culture medium. MC was necessary to limit the movement of single cells or cell clusters after more than 24 hr of observation (*Videos 1 and 5*). We deduce that this seeding protocol has the potential to be effectively employed with various non-adherent cell types. This is based on the principle that cell adhesion on SBS is facilitated by the nanostructure's ability to engage with integrins (*Schulte et al., 2016*), which are widely expressed in diverse cell types (*Johansen et al., 2018*; *Floren et al., 2020*; *Kim et al., 2020*; *Ogana et al., 2024*). The seeding protocol could also certainly be extended to other types of live-cell imaging experiments, especially when the fluorescence intensity can be monitored for shorter durations, thereby circumventing the need for the addition of MC. Moreover, the image and data analysis pipelines can be easily applied to adherent cells, as shown in *Figure 3—figure supplement 1*.

The presented image and data analysis workflows rely on two different software, Fiji and R, widely used by the imaging and data analysis community. The scripts were made publicly available, enabling customization to tailor the workflow according to specific requirements. This not only streamlines the cell cycle analysis of cells expressing the FUCCI(CA)2 indicator, but also makes it accessible to researchers with basic programming knowledge.

In conclusion, we confirm the efficiency of the optimized conditions in immobilizing the non-adherent cells for long time periods, without affecting their predicted response upon different environmental circumstances. The described work allows researchers in the field to analyze thousands of (non-)adherent cells per each experimental condition by using the image and data analysis pipelines

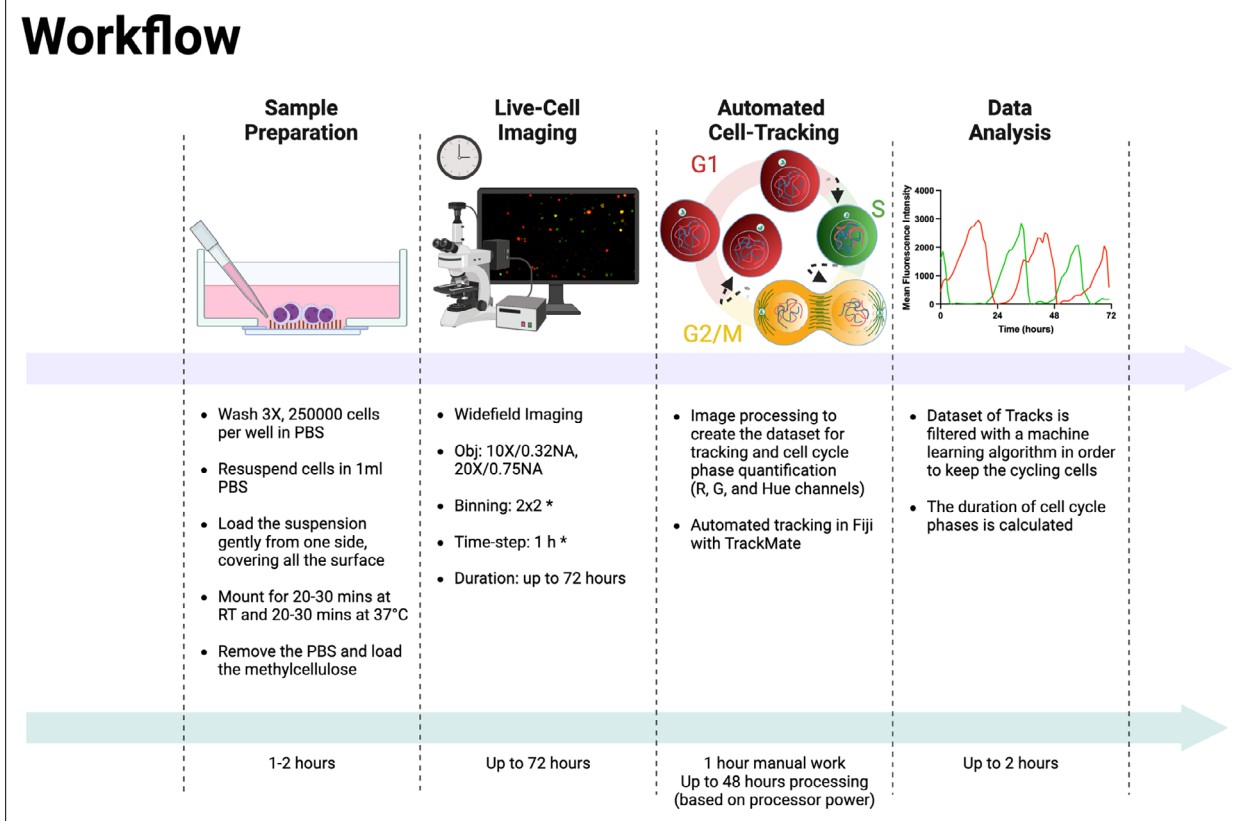

**Figure 5.** Experimental and analysis workflow. Summary of the entire experimental workflow, from cell seeding to the calculation of cell cycle phases. Asterisks in the live-cell imaging section mean optional settings. This figure was created with BioRender.com.

to automatically perform image processing, cell tracking, filtering of incorrect tracks, and cell cycle phases identification and characterization.

## Materials and methods

### Key resources table

| Reagent type (species) or resource | Designation | Source or reference | Identifiers | Additional information |
|---|---|---|---|---|
| Cell line (*Homo sapiens*) | NB4 | DSMZ | ACC 207 | |
| Cell line (*H. sapiens*) | Kasumi-1 | DSMZ | ACC 220 | |
| Cell line (*H. sapiens*) | MDA-MB-231 | ATCC | HTB-26 | |
| Chemical compound, drug | Palbociclib | Selleckchem | S4482 | |
| Chemical compound, drug | Ribociclib | Selleckchem | S7440 | |
| Chemical compound, drug | PF-06873600 | Selleckchem | S8816 | |
| Transfected construct (human) | tFUCCI(CA)2/pCSII-EF | RIKEN BRC through the National BioResource Project of the MEXT, Japan (Cat# RDB15446); Sakaue-Sawano, et al., Mol. Cell 68 (3): 626–640.e5. 2017 | RDB15446 | Lentivirus vector plasmid of mCherry-hCdt1(1/100)Cy(-)_P2A_mVenus-hGem(1/110). |
| Commercial assay or kit | Click-iT EdU Alexa Fluor 647 Flow Cytometry Assay Kit | Thermo Fisher Scientific | C10419 | |
| Commercial assay or kit | Click-iT EdU Cell Proliferation Kit for Imaging, Alexa Fluor 647 dye | Thermo Fisher Scientific | C10340 | |

*Continued on next page*

*Continued*

| Reagent type (species) or resource | Designation | Source or reference | Identifiers | Additional information |
|---|---|---|---|---|
| Commercial assay or kit | TruSeq RNA Library Preparation Kit v2 | Illumina | RS-122-2001 and RS-122-2002 | |
| Software, algorithm | LAS X | Leica Microsystems Inc | https://www.leica-microsystems.com/products/microscope-software/ | |
| Software, algorithm | FIJI | *Schindelin et al., 2012* | https://fiji.sc/ | |
| Software, algorithm | TrackMate | https://doi.org/10.1016/j.ymeth.2016.09.016 | https://imagej.net/plugins/trackmate/ | |
| Software, algorithm | R | *R Development Core Team, 2021* | https://www.r-project.org/about.html | |
| Software, algorithm | Tidyverse | *Wickham et al., 2019* | https://www.tidyverse.org/ | |
| Software, algorithm | Tidymodels | *Kuhn and Wickham, 2020* | https://www.tidymodels.org/ | |
| Software, algorithm | tsfeatures | *Hyndman, 2023* | https://pkg.robjhyndman.com/tsfeatures/ | |
| Software, algorithm | timetk | *Dancho and Vaughan, 2024* | https://business-science.github.io/timetk/ | |
| Other | DAPI stain | Sigma-Aldrich | 32670 | 5 µg/ml |
| Other | Hoechst 33342 stain | Sigma-Aldrich | B2261 | 5 µg/ml |
| Other | Leica Thunder Imager | Leica | https://www.leica-microsystems.com/products/thunder-imaging-systems/ | Widefield microscope |
| Other | DFC9000 GTC | Leica | https://www.leica-microsystems.com/products/microscope-cameras/p/leica-dfc9000/ | sCMOS camera |
| Other | Lumencor Spectra X | Lumencor | https://lumencor.com/products/spectra-x-light-engine | Multi-color illumination |
| Other | Glass-bottom multiwell plate | MatTek Corporation | P12G-1.5-14F | |
| Other | SBS-coated bottom-glass multiwell plates | Tethis S.p.A. | https://tethis-lab.com/contact/ | Titania nanostructured coating |

## Cell culture

### Cell lines and growing conditions

NB4 and Kasumi-1 cells were grown in 3 ml of a mix of 80 ml MethoCult H4230 RPMI-1640 plus 20 ml Roswell Park Memorial Institute (RPMI) 1640 medium with final concentrations of 10% fetal bovine serum (FBS), 2 mM glutamine, and 1% penicillin/streptomycin. MDA-MB-231 cells were grown in Dulbecco's Modified Eagle Medium with 10% FBS + 2 mM L-glutamine and 1% penicillin/streptomycin. All cells were grown according to ATCC recommendations in a humidified tissue culture incubator at 37°C with 5% $CO_2$ environment.

## Generation of cell lines stably expressing FUCCI(CA)2

Cells were put in 24-well plates and plated at a density of 500,000 cells in 500 µl of medium per well. The Lentiviral vector carrying the tFUCCI(CA)2/pCSII-EF (*Sakaue-Sawano et al., 2017*) plasmid was diluted in RPMI 10% serum pen/strep in order to add 500 µl to the cells. Three rounds of infection were carried out in the presence of 5 ug/ml of polybrene (Sigma). Cells were infected simply by adding the 100× concentrated virus supernatant (20–35 µl/well) onto the cells; the plate was centrifuged for 1 hr at room temperature (RT) at 2500 rpm. The medium is added 2 hr post-infection up to 1 ml final. Then it is incubated at 37°C overnight. Infected cells expressed FUCCI(CA)2 probes, emitting mCherry and mVenus. Therefore, the selection of the cells expressing the FUCCI(CA)2 was done by sorting the fluorescence markers, mCherry and mVenus on fluorescence-activated cell sorting (FACS) instrument (FACSAria cell sorter, BD Biosciences, Oxford, UK).

## SBS-coated bottom-glass multiwell plates production

Glass-bottom multiwell plates are coated by Tethis using the Supersonic Cluster Beam Deposition (SCBD) technology (a detailed description of SCBD and its principle of operation can be found in *Piseri et al., 2004*; *Wegner et al., 2006*). A supersonic seeded beam of titania nano-clusters is produced,

under high vacuum, by a pulsed microplasma cluster source (*Piseri et al., 2004*; *Wegner et al., 2006*) and deposited on glass-bottom multiwell plates. The process is tuned to produce nanostructured $TiO_2$ films (thickness from 50 to 200 nm) with a controlled nanoscale morphology. The titania nanostructured coating of the plates is transparent and biocompatible and has a surface topography that promotes the spontaneous adhesion and immobilization of living cells (*Carbone et al., 2006*). The SBS-coated bottom-glass multiwell plates used in this study are available upon request from Tethis S.p.A. (https://tethis-lab.com/contact/).

## Sample preparation

Approximately 250,000 of Kasumi-1 and NB4 cells and 100,000 of MDA-MB-231 cells were plated either in each well of the Tethis SBS 12-well multi-wells or 8-well ibidi plates in triplicates in the presence of the aforementioned compounds at day 0 (T0). Cells were treated with final 2 nM, 10 nM, 50 nM, 250 nM, and 1250 nM of palbociclib; 50 nM, 75 nM, 100 nM, 125 nM, and 150 nM of ribociclib; and 25 nM, 50 nM, 75 nM, 100 nM, and 125 nM of PF-06873600, diluted in advance and directly in the relevant culture medium for each cell line. Only the 50 nM treatments are shown for ribociclib and PF-06873600.

Three cell suspension samples were prepared for triplicate-independent counting and the average of three readings used as the cell count. Cell suspension was diluted with Trypan Blue dye (Sigma) at a 1:1 ratio to identify cell viability using Bio-Rad TC20 automated cell counter.

## RNA extraction and RNA-seq protocol

Total RNA was extracted from dry pallets of cells collected prior and post-acquisitions and purified using the Zymo Research Quick-RNA Miniprep (W/O directzol). Reverse transcription was performed with the SuperScript II Kit (Invitrogen), according to the manufacturer's protocol. RNA-seq was performed according to the True-seq Low sample protocol selecting only polyadenylated transcripts. In brief, before starting mRNA isolation and library preparations, the integrity of the total RNA was evaluated by running samples on a Bioanalyzer instrument by picoRNA Chip (Agilent), then converted into libraries of double-stranded cDNA appropriate for next-generation sequencing on the Illumina platform. The Illumina TruSeq v.2 RNA Sample Preparation Kit was used following the manufacturer's recommendations. Briefly, 0.1–1 µg of total RNA were subjected to two rounds of mRNA purification by denaturing and letting the RNA bind to Poly-T oligo-attached magnetic beads. Then fragmentation was performed exploiting divalent cations contained in the Illumina fragmentation buffer and high temperature. First- and second-strand cDNA is reverse transcribed from fragmented RNA using random hexamers. First-strand cDNA was synthesized by SuperScript II (Invitrogen) reverse transcriptase and random primers and second-strand cDNA synthesized by DNA polymerase I and Rnase H. The subsequent isolation of the cDNA was achieved by using AMPure XP beads (depending on the concentration used, these beads can efficiently recover PCR products of different sizes). The product recovered contained overhanging strands of various lengths due to the fragmentation procedure. The 5′ and 3′ ends of cDNA are repaired by the 3′–5′ exonuclease activity and the polymerase activity and adenylated at 3′ extremities before ligating specific Illumina oligonucleotides adapters followed by 15 cycles of PCR reaction using proprietary Illumina primers mix to enrich the DNA fragments. Prepared libraries were quality checked and quantified using Agilent high-sensitivity DNA assay on a Bioanalyzer 2100 instrument (Agilent Technologies).

## RNA sequencing data analysis

Raw reads 51 bp PE for NB4 and Kasumi-1 cells were quality-filtered and aligned to the hg18 reference genome using nf-core/rnaseq v3.9 pipeline using STAR as aligner and Salmon for quantification with default parameters. Gene counts for each sample were log1p transformed, and mean value among the two replicates was taken to compute Pearson correlation among gene expression pre- and post-time-lapse acquisition.

## Image acquisition

Images were acquired with a Leica Thunder Imager (Leica Microsystems, Wetzlar, Germany), equipped with a Lumencor Spectra X Light Engine (Lumencor, Beaverton, USA) for fluorescence excitation, a motorized stage, and a Leica DFC9000 GTC camera. For non-adherent cells, images were acquired

with LAS X software (Leica Microsystems, version 3.7.5.24914) using a 20×/0.75NA air objective and a binning 2 × 2 was applied to increase the SNR. The mCherry and mVenus signals were detected respectively with 540–580 nm and 460–500 nm excitation filters, 585 and 505 nm dichroic mirrors, and 592–668 nm and 512–542 nm emission filters. The brightfield channel was also acquired for representation purposes. We imaged 20–25 FOVs per well and focal points were manually set in each position before starting the acquisition and kept constant during the whole time lapse thanks to the Adaptive Focus Control (AFC, Leica Microsystems). The total duration of the time lapse on non-adherent cells was 72 hr, and the time interval was set to 1 hr to prevent cell phototoxicity.

Regarding the experiment on adherent cells, a 10×/0.32 NA PH1 dry objective was used and the time-lapse duration was set to 120 hr, with 30 min as timestep. In this case, 10 FOVs per well were acquired.

## Image analysis

The image pre-processing step was performed using a custom-made Fiji macro. Briefly, the pipeline executed a flat-field correction on the fluorescence channels with the plugin Basic (*Peng et al., 2017*), then applied a Gaussian blur (sigma = 1 px), a Top Hat filter (radius = 20 px), and a background subtraction to enhance the cells' signal; duplicated images of red and green channels were merged in a multichannel stack and brightness and contrast values were adjusted, according to each experiment, within the range of 0 to the maximum gray value of the stack's histogram; the multichannel images were then converted into RGBs for each frame, and finally into an HSB stack from which the brightness channel and the hue channel were kept respectively for tracking and cell-phase profiling purposes; the final processed image was saved for further analysis and was composed of four channels: the red and green channels, the brightness channel, and the hue channel.

The tracking analysis was realized through TrackMate (version 7.10.2). We automatized the execution of TrackMate over all the FOVs acquired in the experiment by adapting a Jython script, freely available on the website (https://imagej.net/plugins/trackmate/scripting/scripting). As the shape of our cells was approximately round, the Laplacian of Gaussian (LoG) detector was selected as the cell identifier on the brightness channel, and the Linear Assignment Problem Tracker algorithm was utilized for linking phase, adjusting the Max Linking distance parameter for each experiment. Gap closing was enabled for up to three frames, considering the significant decrease in fluorescence intensity during mitosis and the transition between G1 and S phase, that can eventually affect the detection of a cell, due to the rapid decrease of the red signal and the relatively slow increase of the green one (*Sakaue-Sawano et al., 2017*). Identified tracks were filtered for total duration, track displacement, and starting frame of the track. Specifically, the duration filter maximized the chances to follow at least one cell cycle (track duration > 25 hr), while the cutoff on total displacement assures the deletion of dead cells that are usually motionless (track displacement > 10 px). Tracks that start after the first 10 hr were discarded to avoid tracker errors that may give rise to unreliable cell cycle quantification as the cells tend to form clusters above 24–30 hr of experiment (*Video 1*). The specific values of parameters of the LoG cells' detector (i.e., spot radius, quality threshold) were preliminarily checked on images via the TrackMate GUI, as well as the linking and gap closing max distances and the final filters on tracks' duration, displacement and starting frame. The parameters selected manually were then inserted in the script for the batch execution. As an output, we automatically saved a table of spots for each FOV, corresponding to cells in identified tracks, with the selected TrackMate features (see *Supplementary file 1*).

## Model creation

To assess the traceability of each track in a fast, efficient, and scalable manner, we employed a machine learning approach. A dataset consisting of 2319 manually annotated tracks, with 1939 tracks classified as untraceable and 380 tracks marked as traceable (~1/5), was used. The dataset was divided into training and test sets, with 1855 tracks (~80%) allocated for training and 464 tracks (~20%) reserved for testing while also stratifying for the outcome variable. To address the problem of the unbalanced outcome, a downsampling procedure is employed in the training set.

Time-series-like features associated with each track are extracted (*Wang et al., 2006*; *Hyndman, 2023*), as illustrated in *Figure 2—figure supplement 1A*, displaying the distribution of the 70 features utilized for distinguishing between traceable and untraceable cells.

To optimize model performance and address potential biases of the random forest model employed, we applied stratified tenfold cross-validation on the training set, creating resampling folds. Leveraging these folds, we conducted hyperparameter tuning. Specifically, we explored different values for randomly selected predictors (mtry), number of trees (trees), and minimal node size (min_n) to identify the combination that maximized the area under the receiver operator curve (AUC) (*Sacks et al., 1989*). Subsequently, an additional round of training was conducted using the selected hyperparameters, this time employing the entire training set. The trained model was then evaluated on an independent test set to assess its performance. The results, as depicted in *Figure 2—figure supplement 1B*, revealed an AUC value of 0.971 as well as a sensitivity of 0.897, a specificity of 0.974, and an accuracy of 0.905, confirming the validity of the training procedure. These values were comparable to those obtained from the resampling folds, indicating the absence of overfitting.

Given the relatively small size of the manually annotated training set, we performed an additional and final round of training using the pseudo-labeling framework (*Lee, 2013*). The original model was used to predict unlabeled data, any predicted probability exceeding 0.5 was considered indicative of confidence in the prediction. A new semi-supervised random forest model was trained on this augmented dataset, employing the same hyperparameters as determined previously. This final model achieved an improved AUC of 0.975 on the test set, indicating enhanced track trackability assessment. The schematics of the pipeline, implemented in R (v. 4.3.0) (*R Development Core Team, 2021*) using Tidymodels (v. 1.1.0) (*Kuhn and Wickham, 2020*) and Tidyverse (v.2.0.0) (*Wickham et al., 2019*), can be seen in *Figure 3A*.

## Time-series analysis

To generate time-series-like information, we utilized the abovementioned Fiji pipeline, which provided a table containing intensity values for the red, green, and hue channels. To address missing frames resulting from tracking gaps, an EWMA was fitted to impute these frames.

To further refine the track curves, we applied a two-step smoothing process. Initially, a simple moving average (SMA) was applied, followed by fitting a fixed lambda smoothing spline with $\lambda = 0.0001$. Finally, min-max scaling is used to normalize the tracks within the range of [0, 1] to make the red and green intensity comparable.

To tackle tracking errors introduced by the Fiji pipeline, we applied the aforementioned random forest model to assess the traceability of each cell (*Figure 2B*).

On the tracks classified as trackable, a manual threshold on the hue intensity is employed to determine the cell cycle phases (G1, S, and G2/M). Furthermore, to quantify the cell cycle at the single-cell level, track splitting to isolate individual cell components and perform phase assignment into G1, G1/S, S, and G2/M is performed using a custom R function.

The resulting single-cell tracks enabled the quantification of individual phase durations for each cell. As an example, the cell phase quantification for five different conditions can be observed in *Figure 3*.

## Model deployment

The resulting model was saved as an .rds file to facilitate practical implementation. Moreover, we encapsulated the model as an API in a Docker container using Vetiver (v. 0.2.1) (*Silge, 2023*), enabling easy deployment and usage.

## EdU incorporation and assessment with flow cytometry and imaging

A 2-hr EdU pulse was performed by replacing half of the total media volume of the cells with 2× concentrated EdU in the corresponding growth medium, followed by subsequent fixation. Click-iT EdU Alexa Fluor 647 Flow Cytometry Assay Kit (CN: C10419, Thermo Fisher Scientific, Waltham, MA) and Click-iT EdU Cell Proliferation Kit for Imaging, Alexa Fluor 647 dye (CN: C10340, Thermo Fisher Scientific) were used for flow cytometry and imaging, respectively. The experiments were performed according to the manufacturer's protocols for the mentioned kits.

DNA staining with DAPI using 500 μl of 5 μg/ml DAPI in PBS for $10^6$ cells, followed by overnight incubation at 4°C, was additionally performed for cell cycle profiling by flow cytometry. Alternatively, 5 μg/ml Hoechst 33342 (Thermo Fisher Scientific) was used to stain DNA for imaging purposes.

## Confocal imaging and image analysis

The confocal images were captured using the Eclipse Ti2 microscope (Nikon Europe B.V.) combined with the X-Light V3 spinning disk (CrestOptics S.p.a.), solid-state lasers from the Lumencor Celesta light engine, a multiband dichroic mirror, single-band emission filters, and an sCMOS camera (Kinetix, Teledyne Photometrics). A total of 144 and 464 FOVs were acquired using a PLAN APO $\lambda$ D 60 × 1.42/NA oil immersion objective lens (pixel size of 116 × 116 nm) for the Alexa647 (labeling EdU), mCherry, mVenus, and DAPI signals in Kasumi-1 and MDA-MB-231 cells, respectively.

To assess the mean intensity of each signal within the nuclei, a custom Python-based image segmentation pipeline utilizing the Stardist deep learning algorithm (*Schmidt, 2018*) was employed. The Hoechst 33342 intensity corresponding to the 2N DNA amount was calculated, and cells in the initial S phase were selected based on being EdU positive, having mVenus and mCherry intensities above background, and a DNA content smaller than 1.15 * 2N. All data analysis was conducted using RStudio software.

## Acknowledgements

This work was partially supported by the Italian Ministry of Health with Ricerca Corrente and 5x1000 funds and by the Grant AIRC IG20 to SM. KH was supported by Marie Skłodowska-Curie Innovative Training Network (grant no. 813327 'ChromDesign') and AIRC fellowship (ID: 28467). The authors are thankful to Dr. Marina Mapelli and Michela Bruzzi for their kind help in performing EdU experiments; and to Dr. Mattia Marenda and Giulia Tini and for their kind assistance in image and data analysis.

## Additional information

### Funding

| Funder | Grant reference number | Author |
|---|---|---|
| Ministero della Salute | | Saverio Minucci |
| Fondazione AIRC per la ricerca sul cancro ETS | 28467 | Kourosh Hayatigolkhatmi Saverio Minucci |
| Marie Skłodowska-Curie Innovative Training Network | 10.3030/813327 | Kourosh Hayatigolkhatmi |
| Fondazione AIRC per la ricerca sul cancro ETS | IG20 | Saverio Minucci |

The funders had no role in study design, data collection and interpretation, or the decision to submit the work for publication.

### Author contributions

Kourosh Hayatigolkhatmi, Conceptualization, Formal analysis, Funding acquisition, Validation, Investigation, Visualization, Methodology, Writing – original draft, Writing – review and editing; Chiara Soriani, Emanuel Soda, Data curation, Software, Formal analysis, Validation, Visualization, Methodology, Writing – original draft; Elena Ceccacci, Validation, Methodology; Oualid El Menna, Investigation; Sebastiano Peri, Ivan Negrelli, Giacomo Bertolini, Resources; Gian Martino Franchi, Roberta Carbone, Resources, Methodology; Saverio Minucci, Conceptualization, Resources, Supervision, Funding acquisition, Validation, Methodology, Project administration, Writing – review and editing; Simona Rodighiero, Conceptualization, Supervision, Investigation, Visualization, Methodology, Writing – original draft, Writing – review and editing

### Author ORCIDs

Kourosh Hayatigolkhatmi ⓘ http://orcid.org/0000-0002-9910-9756
Chiara Soriani ⓘ http://orcid.org/0000-0003-4363-6597
Simona Rodighiero ⓘ https://orcid.org/0000-0003-4236-7823

Reviewer #3 (Public review): https://doi.org/10.7554/eLife.94689.3.sa1
Author response https://doi.org/10.7554/eLife.94689.3.sa2

## Additional files

### Supplementary files
MDAR checklist

Supplementary file 1. TrackMate features. This is the table of the spots that is generated after the execution of the tracking script, that automatize TrackMate plugin. For more available parameters and information, please refer to: TrackMate (https://imagej.net/plugins/trackmate/) and Scripting TrackMate (https://imagej.net/plugins/trackmate/scripting/scripting).

### Data availability
The raw datasets related to the plots shown in Figure 3 are publicly available at https://doi.org/10.5061/dryad.cvdncjtcx. A subset of the raw data for the plot shown in Figure 3—figure supplement 1 is available at https://doi.org/10.5061/dryad.cvdncjtcx. The full dataset is available at https://doi.org/10.5061/dryad.tht76hf7w and https://doi.org/10.5061/dryad.9s4mw6mrs. The data table related to the plots in Figure 3—figure supplement 2 can be accessed at https://doi.org/10.5061/dryad.cvdncjtcx. The .fcs files for the plots shown in Figure 4A, 4B, and Figure 4—figure supplement 1A and B are available at https://doi.org/10.5061/dryad.cvdncjtcx.The source code and user manual for the Fiji pipeline is available at https://github.com/ieoresearch/cellcycle-image-analysis (copy archived at *Soriani, 2024*). The source code and user manual for the R pipeline is available at https://github.com/EmanuelSoda/FUCCI_ML (copy archived at *Soda, 2024*).

The following datasets were generated:

| Author(s) | Year | Dataset title | Dataset URL | Database and Identifier |
|---|---|---|---|---|
| Rodighiero S, Ceccacci E, Hayatigolkhatmi K, Soriani C, El Menna O, Soda E | 2024 | Data from: Automated workflow for the cell cycle analysis of (non-)adherent cells using a machine learning approach | https://doi.org/10.5061/dryad.cvdncjtcx | Dryad Digital Repository, 10.5061/dryad.cvdncjtcx |
| Rodighiero S, Hayatigolkhatmi K, Soriani C | 2024 | Fluorescence time-lapse images of MDA-MB-231 cells expressing FUCCI(CA)2 (Part 1/2) | https://doi.org/10.5061/dryad.tht76hf7w | Dryad Digital Repository, 10.5061/dryad.tht76hf7w |
| Rodighiero S, Hayatigolkhatmi K, Soriani C | 2024 | Fluorescence time-lapse images of MDA-MB-231 cells expressing FUCCI(CA)2 (Part 2/2) | https://doi.org/10.5061/dryad.9s4mw6mrs | Dryad Digital Repository, 10.5061/dryad.9s4mw6mrs |

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
