## [Editor Report · eLife assessment]

This article presents a **valuable** automated method to track individual mammalian cells as they progress through the cell cycle using the FUCCI system. The authors have developed a technique for analyzing cells that grow in suspension and used their method to look at different tumor cell lines that grow in suspension and determine the effect of drugs that directly affect the cell cycle. They show **solid** evidence that the method can be applied to both adherent and non-adherent cell lines. This article will be of interest to cell biologists investigating cell cycle effects.

---

## [Referee Report · Reviewer #3 (Public review)]

Summary:

This paper provides presents an automated method to track individual mammalian cells as they progress through the cell cycle using the FUCCI system, and applies the method to look at different tumor cell lines that grow in suspension and determine their cell cycle profile and the effect of drugs that directly affect the cell cycles, on progression through the cell cycle for a 72 hour period.

Strengths:

This is a METHODS paper. The one potentially novel finding is that they can identify cells which are at the G1-S transition by the change in color as one protein starts to go up and the other one goes down, similar to change seen as cells enter G2/M. They have provided detailed data in the resubmission, demonstrating how this can be done in different cell lines and that the resolution of the brief time is about (about 1 hr) when the cells are determined to be in the transition from G1 to S. They further showed how one can explore this period using EDU labeling in conjunction with FUCCI how one can determine whether cells have entered S-phase. This nicely addressed a weakness identified in the previous review.

---

## [Author Response]

The following is the authors’ response to the original reviews.

**Reviewer 1:**
Comment 1 and 2: “The pipeline relies on a large number of hard-coded conditions: size of Gaussian blur (Gaussian should be written in uppercase), values of contrast, size of filters, levels of intensity, etc. Presumably, the authors followed a heuristic approach and tried values of these and concluded that the ones proposed were optimal. A proper sensitivity analysis should be performed. That is, select a range of values of the variables and measure the effect on the output.”“Linked to the previous comments. Other researchers that want to follow the pipeline would have either to have exactly the same acquisition conditions as the manuscript or start playing with values and try to compensate for any difference in their data (cell diameter, fluorescent intensity, etc.) to see if they can match the results of the manuscript.”

We thank the Reviewer for his insightful comments. We have modified the “Usage” section of the GitHub page (https://github.com/ieoresearch/cellcycle-image-analysis) to include, for each step of the image processing, a paragraph explaining the significance of the operation and a paragraph named “Suggested Values Range” where tips for optimal parameter settings are given and examples with different parameter settings are shown. We believe that these new paragraphs help researchers easily customize the pipeline to their own data.

**Reviewer 2:**
Comment 1: “It would be useful to include frames from the movie showing a G1/S cell in Figures 1 and S1 with some indication of how long that cell is present. From Figure S4 it looks like it is substantially less than an hour.It would definitely be nice to validate this observation. A brief pulse of EdU together with the FUCCI colors could allow you to do that in a culture of cycling cells. It appears that the green color as cells enter S-phase develops slowly (and maybe gets brighter continuously) as does the red color as cells progress through G1. It would be nice to validate what the color the cells are when they actually initiate DNA replication.”

We thank the Reviewer for the opportunity to further investigate our results and clarify points that were unclear in the first version of the manuscript. As suggested, we have included all acquired frames depicting the G1 to S transition/early S phase of three cells: the Kasumi-1 untreated cell and the PF-06873600 treated NB4 cell shown in Fig. 1A, and the MDA-MB-231 cell shown in Fig. S1; they are shown in panels D of Fig. 4 and S5, respectively.

For the Kasumi-1 and NB4 cells, the G1 to S transition/early S phase, defined in the pipeline refinement step as a yellow phase appearing before the S phase, is visible at the 12-hour frame. Conversely, the MDA-MB-231 cell shown in Fig. S5D does not exhibit the G1 to S or early S phase, yellow; it transitions abruptly from red to green within our acquisition timeframe (30 min in this case), producing a green early S phase. This observation supports the Reviewer's suggestion that the G1 to S yellow transition is often shorter than one hour and it is not identifiable in all cells.

To further investigate this point, we also conducted the EdU experiments kindly suggested by the Reviewer. Kasumi-1 and MDA-MB-231 cells expressing the FUCCI(CA)2 probes were exposed to a pulse of EdU, and subsequently analyzed using flow cytometry and confocal microscopy. A new paragraph titled “The workflow allows the identification of the G1 to S phase transition” has been added to the Results section, with the corresponding data presented in Fig. 4 and Fig. S5 for Kasumi-1 and MDA-MB-231 cells, respectively. The Methods section has also been updated describing the new experiments.

Additionally, in BOX1 under the 'Cell phase assignment' paragraph, point (III), we have removed point 'a. Re-assign the G2/M frames to G1'. Although theoretically possible according to the pipeline, this reassignment is incorrect in practice because mVenus fluorescence indicates that the cells are starting or have already initiated DNA replication.

All the modifications we made in the text and Figure captions are highlighted in red. We would be thankful if the co-first authorship of Kourosh Hayatigolkhatmi, Chiara Soriani and Emanuel Soda is acknowledged in the final published version of the article.

We believe that the revisions have strengthened our manuscript, and we hope that it now meets the reviewers' suggestions for greater clarity.